# What Are the Key Anatomical Features for the Success of Nose-to-Brain Delivery? A Study of Powder Deposition in 3D-Printed Nasal Casts

**DOI:** 10.3390/pharmaceutics15122661

**Published:** 2023-11-23

**Authors:** Clément Rigaut, Laura Deruyver, Maxime Niesen, Marc Vander Ghinst, Jonathan Goole, Pierre Lambert, Benoit Haut

**Affiliations:** 1Transfers Interfaces and Processes (TIPs), École Polytechnique de Bruxelles, Université Libre de Bruxelles, 1050 Brussels, Belgium; pierre.lambert@ulb.be (P.L.); benoit.haut@ulb.be (B.H.); 2Laboratoire de Pharmacie Galénique et Biopharmacie, Faculté de Pharmacie, Université Libre de Bruxelles, 1050 Brussels, Belgium; laura.deruyver@ulb.be (L.D.); jonathan.goole@ulb.be (J.G.); 3Department of Ear, Nose and Throat and Cervico-Facial Surgery, CUB Hôpital Erasme, Hôpital de Bruxelles (HUB), 1070 Brussels, Belgium; maxime.niesen@ulb.be (M.N.); marc.van.der.ghinst@ulb.be (M.V.G.)

**Keywords:** 3D printing, nasal cast, nose-to-brain, clustering, olfactory region, instillation, blood–brain barrier, neurological diseases

## Abstract

Nose-to-brain delivery is a promising way to improve the treatment of central nervous system disorders, as it allows the bypassing of the blood–brain barrier. However, it is still largely unknown how the anatomy of the nose can influence the treatment outcome. In this work, we used 3D printing to produce nasal replicas based on 11 different CT scans presenting various anatomical features. Then, for each anatomy and using the Design of Experiments methodology, we characterised the amount of a powder deposited in the olfactory region of the replica as a function of multiple parameters (choice of the nostril, device, orientation angle, and the presence or not of a concomitant inspiration flow). We found that, for each anatomy, the maximum amount of powder that can be deposited in the olfactory region is directly proportional to the total area of this region. More precisely, the results show that, whatever the instillation strategy, if the total area of the olfactory region is below 1500 mm^2^, no more than 25% of an instilled powder can reach this region. On the other hand, if the total area of the olfactory region is above 3000 mm^2^, the deposition efficiency reaches 50% with the optimal choice of parameters, whatever the other anatomical characteristics of the nasal cavity. Finally, if the relative difference between the areas of the two sides of the internal nasal valve is larger than 20%, it becomes important to carefully choose the side of instillation. This work, by predicting the amount of powder reaching the olfactory region, provides a tool to evaluate the adequacy of nose-to-brain treatment for a given patient. While the conclusions should be confirmed via in vivo studies, it is a first step towards personalised treatment of neurological pathologies.

## 1. Introduction

The interest in the nasal cavity as a systemic or targeted administration route has grown over the last few years. The common goal of each recent nasal spray for this purpose is the rapidity of action, see, for instance, Zavzpret^TM^ [1] and Trudhesa^®^ [2] for the migraine crisis and Opvee^®^ [3] and Narcan^®^ [4] as rescue medicine for opioid overdose. Notably, the nasal route allows reaching the brain directly, bypassing the blood–brain barrier (BBB) via the olfactory nerves (Figure 1), the trigeminal nerves [5] or the nasal lymphatic pathway [6]. This is commonly referred to as nose-to-brain delivery (N2B) [5,7,8]. Another substantial point is that the nasal cavity is highly vascularised, which increases drug absorption [9,10,11].

Unfortunately, there are many shortcomings in the guidance for nasal products. Specific international regulations for nasal medication products do not exist [13]. The European Medicines Agency (EMA) proposes a guideline on the pharmaceutical quality of inhalation and nasal products (EMEA/CHMP/QWP/49313/2005 [14]). However, it only contains recommendations about particle or droplet size distribution analysis via dynamic light scattering (DLS). Concerning the other quality guidance (e.g., dose uniformity, physical characteristics and delivered dose uniformity), the EMA refers to the oral inhalation products guidance. For inhalation powder products, the guidance recommends using a multistage impactor to determine the particle size distribution. This size distribution can then be linked to the powder proportion in each part of the lungs [15]. Since such a standard tool does not exist for the nose, there is a need for a new tool for in vitro testing of nasal products: the so-called nasal cast, a 3D-printed reproduction of a nasal cavity.

Three-dimensional printing is a great tool to study in vitro an organ before in vivo studies. The 3D-printed replicate of an organ provides more information than a simple 3D visualisation. For instance, Sulaiman et al. reproduced precisely the internal and the external anatomy of an aortic aneurysm by using 3D-printed technology after three-dimensional magnetic resonance angiography on a real patient. This replicate mimics the elastic properties of vessels and constitutes an in vitro model of an aortic arch aneurysm for endovascular procedure simulation [16].

In this work, we used 3D-printing technology to replicate specific nasal anatomies derived from real patients. Despite the difference from a real nasal cavity due to the absence of the mucociliary clearance and enzymatic degradation, a 3D-printed nasal cast is a great tool for an in vitro study. Indeed, such a patient-specific nasal cast can be used to generate a deposition cartography that could be interesting for comparing generics or helping to choose pharmaceutical forms, administration devices, or personalised administration procedures for aiming the zone of interest in the nasal cavity.

A crucial point regarding the nose is that each person has their own anatomy and thus an individual optimal nasal deposition for a given product. The shape of the nostrils and the morphology of the airways are variations that give each person a distinct nose anatomy [17,18]. For instance, a symmetric nasal cavity is extremely rare in the population, and several nasal pathologies are frequent. For example, the principal causes of nasal airway obstruction are the deviation of the septum and the hypertrophy of the inferior turbinates [19]. Nasal septal deviation (NSD) is an extremely common anatomical pathology in the world’s population. Indeed, Mladina et al. studied a population of 2589 adults and found an NSD for 89% of the subjects [20]. Clark et al. found a prevalence of 72% for septal deviation in the studied patients with nasal airway obstruction [19]. Trocino et al. analysed the digital panoramic radiographs of 516 patients to determine the alterations and pathologies prevalence and found an incidence of 59% for nasal turbinate hypertrophy [21]. We can also highlight that septum perforation has a prevalence of around 1% [22,23]. In addition to these pathologies, we can mention that the sinonasal anatomy of children differs significantly from the one of adults. Ertekin et al. studied the development of the nasal cavity and turbinates according to age and sex. They demonstrated that the morphology of the nasal cavity evolves in both sexes until 15 years old [24].

Despite these variations in the morphology of the nose, no study has discussed the difference between patients. Indeed, most of the studies have implied only one anatomy and have varied only the administration parameters [25,26,27]. On the other hand, studies including multiple anatomies do not explain the origin of the differences in the results. For instance, Warnken et al. determined the individual optimal angle for a series of patients [28] but did not relate each angle to an anatomical characteristic.

The general objective of this in vitro work is to study the influence of the anatomy of the nasal cavity on the deposition of a powder in the olfactory zone. Indeed, the N2B delivery is based on the targeting of the olfactory region, which corresponds to around 10% [29,30] of the entire nasal cavity and is difficult to reach due to the tortuous shape of the passages. Failure to provide a sufficient dose of a drug to this region prevents effective N2B treatment.

For this purpose, we used 3D printing to produce nasal casts from 11 different CT scans presenting various anatomical features. Then, for each anatomy and using the Design of Experiments methodology, we characterised the amount of powder deposited in the olfactory region of the cast as a function of multiple parameters (choice of the nostril, device, orientation angle, and the presence or not of a concomitant inspiration flow).

We focused on the powder form due to its numerous advantages intended for N2B delivery in comparison to the liquid form. It can increase the drug bioavailability by increasing the residence time and decreasing the mucociliary clearance, which are the two most important barriers to nasal delivery. Moreover, it is easier to control the size of the particles of a powder, which is an important factor in reaching the olfactory region. Finally, the powder form shows greater stability during storage and avoids adding preservatives [31,32,33].

Finally, our goal is to analyse the results to generate a classification that depends on the anatomical characteristics of the patients. This clustering could predict the outcome of nose-to-brain treatment for each patient. So, it would also be a tool to choose patient-specific treatment, whether via an appropriate instillation procedure or via a complete change of the administration method.

## 2. Materials and Methods

### 2.1. Nasal Cast

#### 2.1.1. Choice of the Patients

For this study, we selected eleven patients with various anatomical properties in terms of their nasal cavity. We choose one standard anatomy (C1), three normal anatomies (C2, C3 and C4), one anatomy showing septum perforation (C5), two paediatric anatomies (C6 and C7), two anatomies showing septum deviation (C8 and C9), and two anatomies showing hypertrophy of the turbinates (C10 and C11). Table 1 summarises the different anatomies and their characteristics (age and sex). All these anatomies derive from CT scans of patients at the Erasme Hospital (Brussels, Belgium), except for the standard anatomy (C1). This anatomy was created by Liu et al. based on the CT scans of 30 healthy patients to generate a median nasal geometry [34]. The type of nasal pathology was attested by ear, nose, and throat (ENT) specialists at the Erasme Hospital. The coronal slice of each anatomy used in this study is provided in Appendix A.

#### 2.1.2. Creation of 3D-Printed Nasal Casts from the CT Scans

We used the procedure described by Rigaut et al. to generate a 3D-printed nasal cast from each of the 11 CT scans [35]. The software used for this procedure were InVesalius (v. 3.1.1; Centro de Tecnologia da Informação Renato Archer, Campinas, Brazil), Meshlab (v. 2022.02; Istituto di Scienza e Tecnologie dell Informazione, Pisa, Italy), and FreeCAD (v. 0.21.1). We cut the final 3D geometry of the nose into six different parts (i.e., the nostrils, the olfactory region, the middle turbinate region, the lower turbinate region, the nasopharynx, and the postnasal fraction). We printed each part with a Form3 printer (Formlabs, Somerville, MA, USA) using Formlabs’ Black Resin. Each part was printed with a resolution of 50 microns. Figure 2 shows the nasal cast C3.

### 2.2. Methods

#### 2.2.1. Design of Experiments

We constructed a Design of Experiments (DoE) for each nasal cast based on four parameters (Table 2) [36]. The first parameter was the instillation device. We used three devices: two unidirectional devices (Aptar UDS and Miat insufflator, see Figure 3a,c) and one bidirectional device (IPMed TriVair^TM^, see Figure 3b). The second parameter was the instillation angle. We chose two strategies: an instillation angle directly aiming at the olfactory region (i.e., a direct line between the exit of the device and the centre of the olfactory region) and another one aiming at the centre of the nasal valve. Third, we varied the instillation side: left nostril or right nostril. Finally, we considered a possible inspiratory flow concomitant to the instillation. Three situations were defined: without inspiratory flow (0 L/min), normal inspiratory flow (15 L/min), and sniff condition (60 L/min). The DoE for each anatomy was performed with the Design-Expert^®^ software (Version 13, Stat-Ease Inc., Minneapolis, MN, USA). The number of runs varied between 15 and 30, depending on the number of interactions. The criterion chosen for determining the run number was to obtain a power of at least 80% for each DoE. The selected design was the factorial randomised optimal design. We interpreted the results of each DoE by fitting a linear model using the four parameters as the input and the ratio of the mass of powder deposited in the olfactory region to the mass of powder injected in the cast as the output. This model is hereafter referred to as the “predictive deposition model”. We kept only the parameters with a significant effect on the olfactory deposition when using analysis of variance (ANOVA). A *p*-value < 0.05 was considered statically significant.

#### 2.2.2. Deposition Tests

We used a deposition test protocol described in our previous works [35,37]. For each experiment, we first coated each part of the nasal cast with an artificial mucus (5% *w*/*w* of Poloxamer^®^ 407 and solution in Simulated Nasal Electrolyte Solution (SNES)). We used Poloxamer^®^ 407 to generate a thermosensitive gel (it is liquid under 10 °C and it swells at around 18 °C) to obtain a thin adherent layer of mucus on the nasal cast.

Then, each part of the nasal cast was assembled before the powder instillation. In the case where an inspiratory flow was needed, it was fixed by using a DFM3 flow meter (Copley Scientific, Nottingham, UK) and was produced with two HCP5 air pumps (Copley Scientific) connected in series to a TPK critical flow controller (Copley Scientific). We used a steady flow to realise the experiments. For that, we turned on the pumps at least 5 s before the instillation.

The mass of the powder introduced in the instillation device was fixed at 25 mg for all the experiments, which is within the acceptable range for nasal delivery [33,38]. We used caffeine as a model powder. The caffeine was sifted through a 0.123 mm sieve to deagglomerate and obtain a particle diameter closer to the ideal size for N2B delivery (around 12 μm [39]). A summary of the powder characteristics at the exit of each device is shown in Table 3. We weighted the device before and after the instillation to control the exact mass of powder injected into the nasal cast. The experiment was repeated if less than 70% of the powder was injected into the cast. The insertion angle of the device was controlled using 3D-printed supports, ensuring the correct spray location and angle. Concerning the unidirectional devices, the performance of the actuation is patient-independent. Thus, we realised its actuation by hand. The actuation of the bidirectional device was accomplished with a blower bulb to mimic an expiration (140 ± 23 L/min) [40].

Finally, we disassembled the nasal cast and rinsed separately each of its parts with ethanol absolute. For each part of the cast, the concentration of caffeine in the resulting solution was measured at 274 nm using UV spectrophotometry (Implen NanoPhotometer^®^) [35] and converted into the mass of the instilled powder deposited in the part. All the experimental results are available in Appendix A. Hereafter, the ratio of the mass of powder deposited in the olfactory region to the mass of powder injected in the cast is referred to as the “olfactory deposition” (expressed in %).

### 2.3. Cast Classification

#### 2.3.1. Geometrical Characterisation of the Anatomies

The first step taken to compare the anatomies between them was to measure the anatomical elements that could impact the deposition of the powder in the olfactory region. The two most prominent features are the olfactory zone and the nasal valve, which restrains access to the turbinates. For each anatomy, we measured the area of each side of the internal nasal valve (the “internal” is dropped hereafter) and its total area (i.e., the sum of the areas of its two sides). For the olfactory region, we measured the length, width, depth, and area of its two sides and its total area (i.e., the sum of the areas of its two sides). We also measured the area and the volume of the whole nasal cavity, as well as the length, width, and depth of its minimum bounding box, to ensure that we missed no valuable variable or interaction.

#### 2.3.2. Correlation between Anatomy and Deposition

We then combined the anatomical measurements and the results of the deposition tests to expose the link between the anatomy and the deposition. This link is essential to switch from individual to general observations. For this, we plotted a correlation matrix [41] between the anatomical measurements and the parameters of the predictive deposition model (i.e., the linear model fitting the deposition results, with the administration parameters as input and the expected olfactory deposition as output). The correlation matrix is a 2D representation of the Pearson correlation coefficient between the variables on the vertical axis and the variables on the horizontal axis. In this way, the relations between the variables are easier to spot and can be used to predict, for instance, the olfactory deposition given a set of anatomical measurements.

#### 2.3.3. Anatomy Clustering

To confirm mathematically the relationships highlighted by the correlation matrix, we represented each anatomy by its predictive deposition model obtained in Section 2.2.1, and we used exploratory factor analysis to identify the variables influencing the olfactory deposition. If a parameter was not statistically significant for an anatomy, its value was set to 0 in the analysis. We used the elbow of the scree plot to determine the trade-off between a small number of factors and a good data representation. We then linked the anatomies with hierarchical clustering using Ward’s method. We determined the final number of clusters via the elbow method applied to the Euclidian distance between the newly grouped points in the factorial space. We finally applied k-means clustering, with the number of clusters defined via the hierarchical method. The resulting points of the hierarchical clusters were used as seeds for k-means algorithm initialisation to ensure the stability of the clustering. A detailed explanation of the techniques used can be found in Appendix A.

## 3. Results and Discussion

### 3.1. Individual Optimisation of the Delivery in the Olfactory Region

As mentioned previously, after the statistical analysis of the DoE, the Design-Expert^®^ software (Version 13) can generate, for each cast, a predictive model (the so-called predictive deposition model), which notably allows us to determine the values of the parameters allowing us to maximise the olfactory deposition (yielding what we call the “maximal olfactory deposition”) or to determine the mean value of the olfactory deposition for the whole set of parameter values. The robustness of this predictive model was validated in a previous study [35]. The values of the parameters allowing us to obtain the maximal olfactory deposition for a given cast are referred to hereafter as the “optimal parameters” for this cast. Figure 4 illustrates the difference between the mean and the maximal olfactory deposition obtained with the predictive deposition model constructed for each nasal cast. The results show that each nasal cast has a significantly better olfactory deposition when using the optimal parameters than the mean olfactory deposition, although use of another method to optimise this olfactory deposition would be interesting. Indeed, the experimental work to construct the different predictive deposition models is laborious and time-consuming.

### 3.2. Classification by Anatomical Trait

The nasal cavities can be referred to by several anatomical traits, such as septal deviation, septal perforation, hypertrophy of the inferior turbinates or age (child versus adult). As said earlier, the principal causes of nasal obstruction are septal deviation and hypertrophy of the inferior turbinates. Consequently, these anatomical traits could have an influence on drug deposition in the nasal cavity due to the obstruction they cause [20,42,43]. Indeed, several studies in the literature highlight the influence of these anatomical traits on the olfactory deposition. A study by Frank et al. concluded an approximately four times less post-nasal-valve deposition on the obstructed size [44]. Another study by Hosseini et al. analysed the regional nasal deposition of MAD Nasal^TM^ in nasal replicas of an adult, a child, and a toddler. They showed a significant difference in the deposition pattern between the adult and the child replicas but no significant difference between the child and the toddler replicas [45].

Thus, we could expect to be able to classify the nasal casts by their anatomical trait and thus to obtain a correlation with the olfactory deposition. But, as illustrated in Figure 5, the results show no correlation between the anatomical trait and the maximal olfactory deposition. Indeed, we see first that the difference between the maximal olfactory depositions of the two paediatric casts is 37%. Moreover, the three normal anatomies have significantly different maximal olfactory depositions (36, 48, and 22%) and the two anatomies with a septal deviation show a difference of 10% in the maximal olfactory deposition.

As a conclusion, the results obtained for the olfactory deposition in the different anatomies do not seem to have a connection with the anatomical traits. It is thus necessary to identify the key parameters responsible for these differences in olfactory deposition.

### 3.3. Correlation between the Anatomy and the Predictive Deposition Model

Figure 6 is a correlation matrix representing, on the horizontal axis, the coefficients of the predictive deposition model, the maximal olfactory deposition predicted by this model (opt) and the absolute difference between the maximal olfactory deposition obtained with an instillation by the right nostril and the one obtained with an instillation by the left nostril (dopt, also predicted by the model) and, on the vertical axis, the different anatomical measurements. A summary of the anatomical measurements for each anatomy is given in Table 4. In Figure 6, the lighter the colour is, the lower the correlation is. A green colour indicates a positive correlation, and a pink colour indicates a negative correlation. From this figure, we can see that the total area of the nasal valve and the overall dimensions of the nasal cavity do not play a significant role in the efficiency of the olfactory deposition, given their low correlations with the coefficients of the predictive deposition model. On the other hand, we see in Figure 6 that the total area of the olfactory region (olf) correlates positively with parameters I and A1 and with the maximal olfactory deposition (opt). I is the intercept of the deposition model. In other words, it is linked to the mean deposition in the olfactory region when confounding all the levels for all the parameters. Thus, a higher proportion of the spray deposits in the olfactory region of the anatomy if I is high. The other coefficient correlating with the total area of the olfactory region, A1, represents the difference between the UDS and the TriVair^TM^; A2 being the difference between the TriVair^TM^ and the Miat insufflator. So, these coefficients reveal how much the olfactory deposition can be improved by using the most appropriate delivery device. The link between the maximal olfactory deposition and the coefficients A1 and I can be understood as follows: an anatomy having a high mean performance (high I) that can be further improved using the UDS (high A1) logically has a high maximal olfactory deposition. The total area of the olfactory region is thus an indicator of the maximal deposit achievable in each anatomy. This conclusion was already outlined as a hypothesis of our previous work [35] based on only two patients, although the current study confirms it with a broader sample.

We see also in Figure 6 that the relative difference between the areas of the left and right sides of the nasal valve correlates strongly with the absolute difference in the maximal olfactory deposition between the two sides (dopt), D and its interaction with the other factors. On the other hand, we can see in Figure 6 that the total area of the nasal valve itself does not play a role in the maximal olfactory deposition. It is probably linked to the narrow plume angle of all our devices, as already exposed in our previous study [35]. However, if one side of the nasal valve is too narrow, it becomes an obstacle for the powder and generates asymmetry in the olfactory deposition.

### 3.4. Deposition Efficiency Prediction

Once the correlations have been drawn, we can try to predict the deposition in the olfactory zone on the sole basis of the anatomical measurements made previously. For this, we first represent the maximal olfactory deposition (i.e., the deposition given by the predictive deposition model using the optimal parameters) versus the total area of the olfactory region (Figure 7a). A linear trend can be seen, confirming the correlation between the two values.

Similarly, we can represent the absolute difference between the maximal olfactory deposition obtained with an instillation by the right nostril and the one obtained with an instillation by the left nostril (dopt) versus the relative difference between the areas of the left and right sides of the nasal valve (Figure 7b). We see here a piecewise linear trend, with dopt close to zero if the relative difference between the areas of the left and right sides of the nasal valve is lower than 20%, and then a linear increase in dopt with this relative difference if it is above 20%. We see also that the two casts presenting the biggest asymmetry in deposition are the two most asymmetrical casts (C3 and C9, as shown in Table 4).

While an exact olfactory deposition prediction is beyond the scope of these graphs, they still provide handy information about the outcome of a potential nose-to-brain treatment. Indeed, they show that, whatever the instillation strategy (notably the choice of the device), if the total area of the olfactory region is below 1450 mm^2^, no more than 25% of an instilled powder can reach this region. On the other hand, if the total area of the olfactory region is above 3000 mm^2^, the deposition efficiency reaches 50% with the optimal choice of the parameters, whatever the other anatomical characteristics of the nasal cavity. Moreover, if the relative difference between the areas of the two sides of the nasal valve is less than 20%, it turns out that there is little interest in the choice of the nostril since the expected absolute difference in the maximal olfactory deposition between left and right nostrils is negligible. On the other hand, the more pronounced the relative difference between these two areas is, the more critical it is to choose the nostril (with dopt larger than 25% if the relative difference between the areas of the two sides of the nasal valve is larger than 41%).

### 3.5. Exploratory Factor Analysis and Clustering

To confirm these conclusions and ensure no substantial latent variable has been missed, we performed an exploratory factor analysis of all the anatomies, based only on their associated predictive deposition model. The results of this analysis show that the variance explained by the first two factors is 39% and 22% of the total variance, respectively, while the third factor explains only 12%. In other words, the first two factors explain more than 60% of the variability between the casts, and adding a third factor improves this by only 10%. It supports thus the result obtained previously: there are only two main anatomical measurements influencing the olfactory deposition: the total area of the olfactory region and the relative difference between the areas of the two sides of the nasal valve.

Table 5 presents the mean and maximal deposition for each cast, along with the parameters for reaching this optimal deposition. Figure 8 shows the results of this analysis with two factors and the subsequent clustering. A clear trend appears where the first factor (horizontal) separates casts C3 and C9 from the others. These two casts are the most asymmetrical ones, with a relative difference between the areas of the two sides of the nasal valve of 45% for C3 and of 42% for C9 (see Table 4). This leads to an absolute difference between the maximal olfactory deposition obtained with an instillation by the right nostril and the one obtained with an instillation by the left nostril of 31% and 35%, respectively (see Figure 7b). While other casts (C5, C10 and C11) also present an asymmetry (see Table 4), the maximal difference between the optimal deposition in the two nostrils for these casts is 16% (cast C5, see Figure 7b).

When integrating the second factor, the maximisation of inter-cluster variance gives rise to four clusters: one comprising the asymmetric casts (C3 and C9), for which the choice of the instillation nostril is critical, and three “side-independent” clusters, for which the nostril chosen for the spray is less crucial (dvalve < 41%, see Figure 7b). The three “side-independent” clusters differentiate with the proportion of the instilled powder reaching their olfactory region. First, C6 shows a very high performance, both for the mean and maximal olfactory deposition (see Figure 4 and Table 5). Second, C7, C10 and C11 show the poorest performance, which can be improved by changing the device, albeit without reaching the performance of the other clusters (see Figure 4 and Table 5). Third, the last cluster (e.g., C1, C2, C4, C5 and C8) gathers the casts with a medium olfactory deposition that can still be improved by using the UDS device (see Figure 4 and Table 5). The mean olfactory deposition, as calculated by the predictive deposition model at a fixed instillation device, confirms this conjecture: for C6, it is equal to 47% with the UDS, 20% with the Miat insufflator, and 24% with the TriVair; while for C10, it is equal to 15% with the UDS, 12% with the Miat, and 14% with the TriVair. In a nutshell, our analysis shows that the anatomies grouped by their mean olfactory deposition also share the same improvement potential. In other words, the three “side-independent” clusters define the suitability of nose-to-brain delivery for each anatomy since a cast that has low mean olfactory deposition has also a low maximal deposition (see Table 5).

## 4. Conclusions

We performed a broad study of powder spray deposition in 11 nasal casts presenting various anatomical features, with a focus on the deposition in the olfactory region. We found out that there are only two main anatomical measurements influencing the olfactory deposition: the total area of the olfactory region and the relative difference between the areas of the two sides of the internal nasal valve.

First, the total area of the olfactory region is proportional to the amount of powder that can be deposited there. Less than 25% of the spray particles are expected to reach the olfactory zone of patients with a total olfactory area of less than 1450 mm^2^. This indicates an incompatibility of a patient with a small olfactory area and nose-to-brain delivery. On the other hand, if the total area of the olfactory region is above 3000 mm^2^, the deposition efficiency reaches 50% with the optimal choice of the parameters.

Second, the relative difference between the areas of the left and right sides of the internal nasal valve is strongly linked to the deposition difference between the two sides of the olfactory region. If the relative difference between the areas of the two sides of the nasal valve is less than 20%, it turns out that there is little interest in the choice of the nostril since the expected difference in maximal olfactory deposition between left and right nostrils is negligible. On the other hand, if it is above 40%, the more critical it is to choose the nostril, as the difference in deposition between both sides can be higher than 25% (absolute deposition).

In this work, we used a powder spray instead of a liquid one. We can expect variations in the deposition between these two forms because of the difference in the particle–wall interaction. On the one hand, it is known that powder particles rebound off the walls [46], which leads to a more posterior deposition. On the other hand, according to the spray–wall model of Kolanjiyil et al., using water particles of 20 μm and a spray velocity between 5 and 20 m/s, the droplet would spread on the surface without rebounding [47]. This would lead to a more anterior deposition and, so, seems less appropriate to reach the olfactory region.

The present work highlights for the first time a correlation between anatomical measurements and the powder deposition in the olfactory zone. As such, it allows for predicting if N2B treatment would be appropriate for a given patient and if the choice of the nostril for the administration is important. Given the differences between biological and nasal cavities and 3D-printed replicas (such as the lack of mucociliary clearance), these conclusions should be confirmed via in vivo studies.

## Figures and Tables

**Figure 1 pharmaceutics-15-02661-f001:**
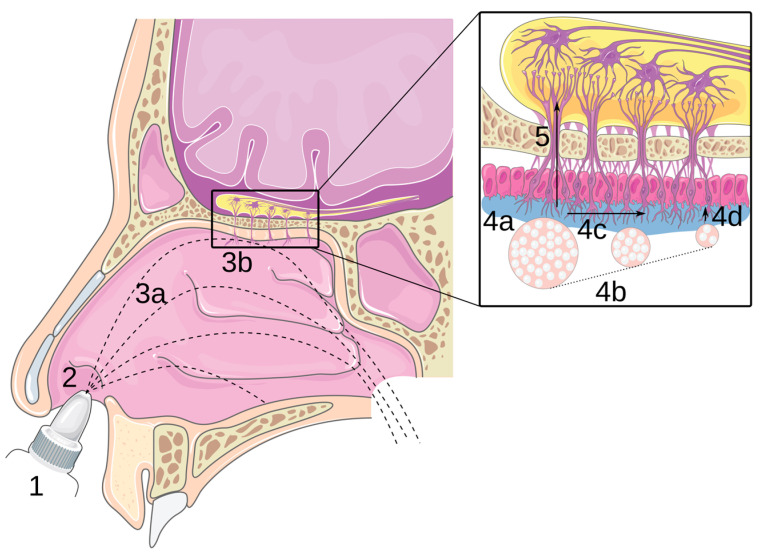
Principle of N2B delivery: (1) drug formulation; (2) instillation; (3a,3b) transport in the nasal cavity and impaction of the mucosa; (4a–4d) transport in the olfactory mucosa: adhesion, dissolution, mucociliary clearance and diffusion; and (5) transport through the epithelium and along the olfactory nerve. Adapted with permission from [12] Copyright 2021 Elsevier.

**Figure 2 pharmaceutics-15-02661-f002:**
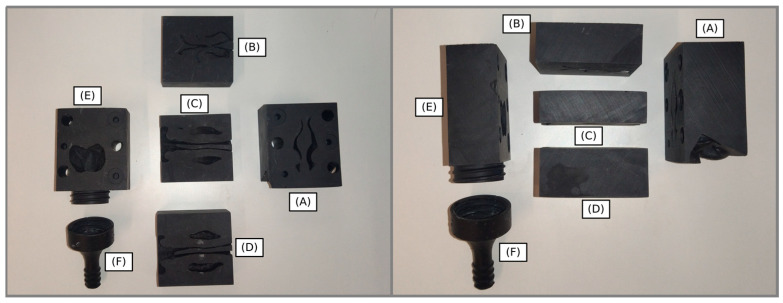
3D-printed nasal cast C3: (A) nostril, (B) olfactory region, (C) middle turbinate region, (D) lower turbinate region, (E) nasopharynx, and (F) post-nasal region.

**Figure 3 pharmaceutics-15-02661-f003:**
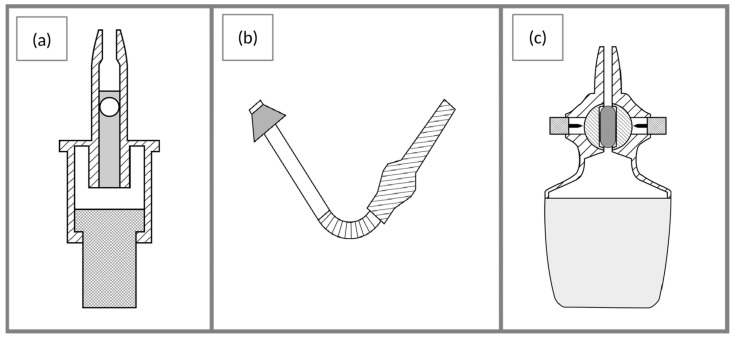
Schematic representation of each device: (**a**) unidirectional device, Unidose System (UDS), Aptar, Le Vaudreuil, France; (**b**) bidirectional device, TriVair^TM^, IP Med Inc. Oceanside, NY, USA; and (**c**) unidirectional device insufflator, MIAT.

**Figure 4 pharmaceutics-15-02661-f004:**
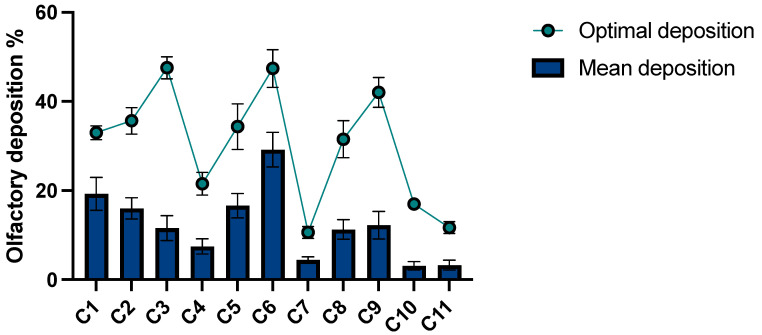
Mean and maximal olfactory deposition for each anatomy. The data are expressed as the mean ± standard error of the model.

**Figure 5 pharmaceutics-15-02661-f005:**
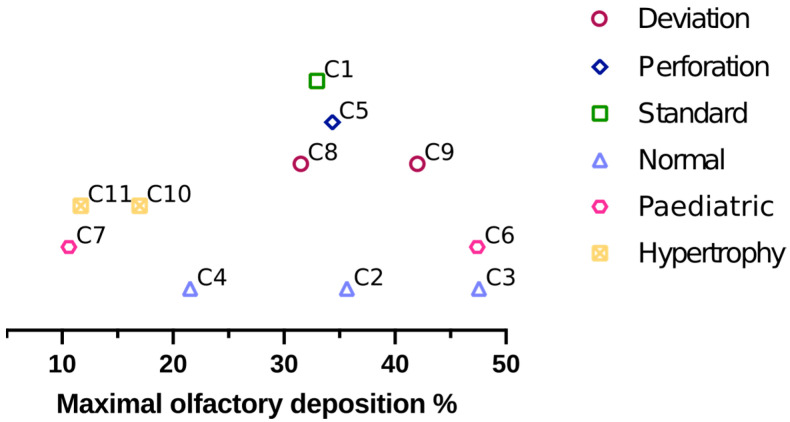
Maximal olfactory deposition for each cast, grouped by their anatomical trait (septum deviation, septum perforation, standard cast, heathy nose, paediatric cavity, and hypertrophy of the inferior turbinates).

**Figure 6 pharmaceutics-15-02661-f006:**
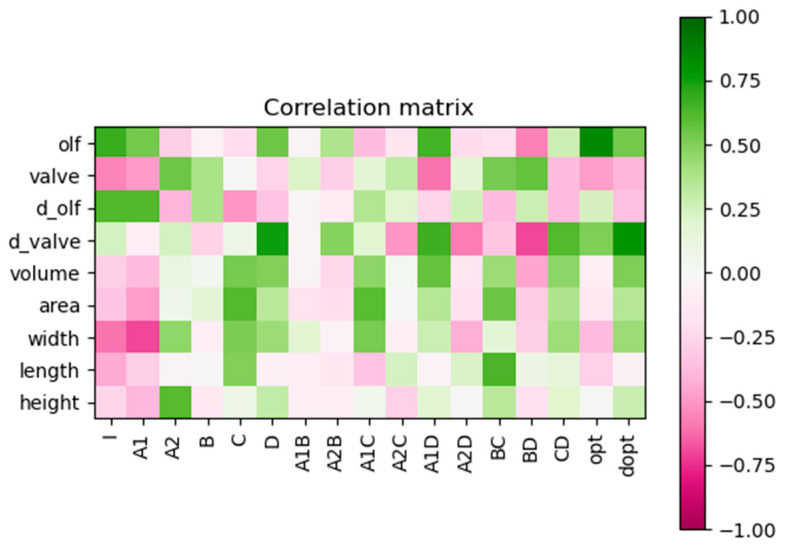
Correlation matrix between, on the horizontal axis, the predictive deposition model coefficients (I, A1, A2, B, C, D, A1B, A2B, A1C, A2C, A1D, A2D, BC, BD, CD), the maximal olfactory deposition predicted by this model (opt), and the absolute difference between the maximal olfactory deposition obtained with an instillation by the right nostril and the one obtained with an instillation by the left nostril (dopt, also predicted by the model) and, on the vertical axis, the anatomical measurements. “olf” is the total area of the olfactory region, “valve” is the total area of the nasal valve, “d_olf” is the relative difference between the areas of the left and right sides of the olfactory region, “d_valve” is the relative difference between the areas of the left and right sides of the nasal valve, “volume” is the overall volume of the nasal cavity, “area” is the overall area of the nasal cavity, “width” is the width of the bounding box, “length” is the length of the bounding box, and “height” is the height of the bounding box.

**Figure 7 pharmaceutics-15-02661-f007:**
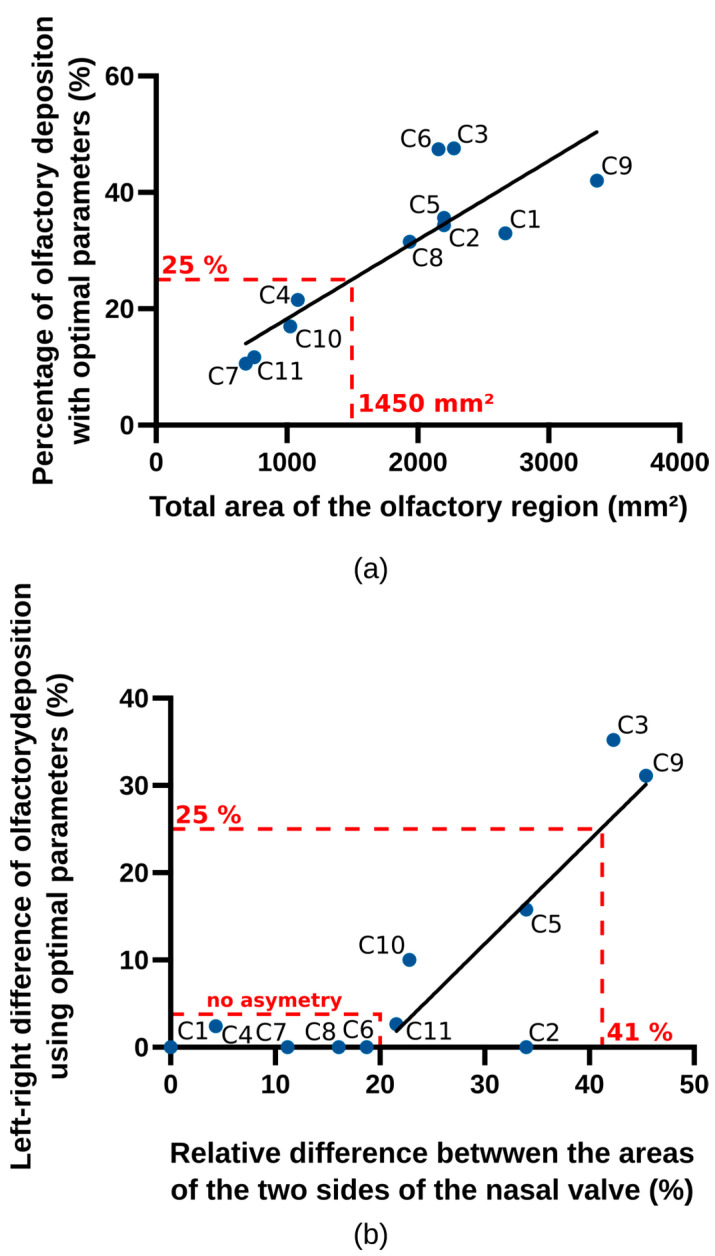
(**a**) Maximal olfactory deposition versus the total area of the olfactory region. (**b**) Absolute difference between the maximal olfactory deposition obtained with an instillation by the right nostril and the one obtained with an instillation by the left nostril (dopt) versus the relative difference between the areas of the two sides of the nasal valve.

**Figure 8 pharmaceutics-15-02661-f008:**
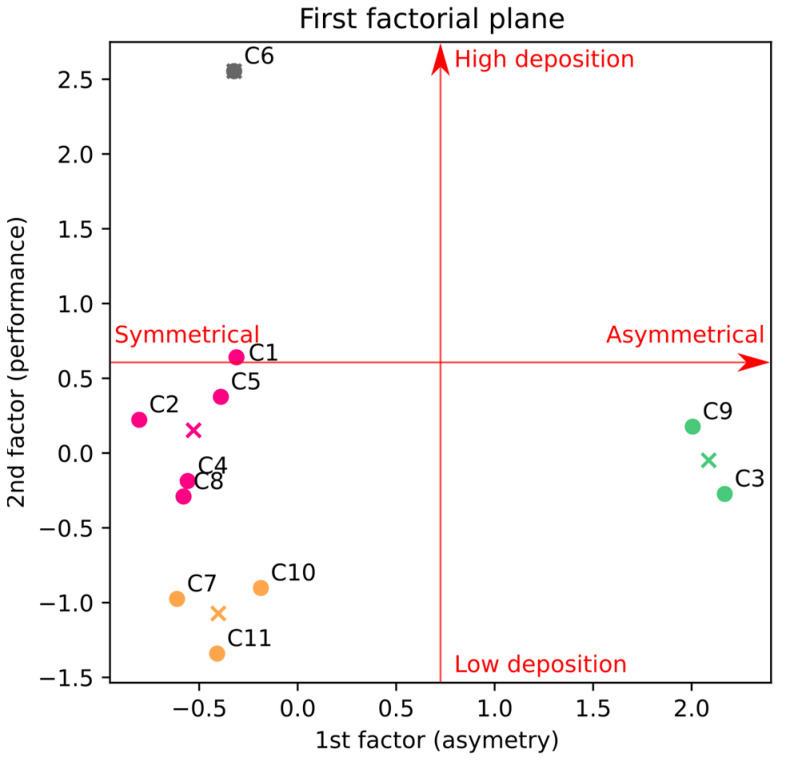
Position of the casts in the factorial plane resulting from exploratory variable analysis and clustering using k-means. The crosses represent the centroid of the clusters.

**Table 1 pharmaceutics-15-02661-t001:** Summary of the different anatomies. * males (M) and females (F).

Anatomy	Sex (M/F) *	Age (Years)	Cast
Standard	-	-	C1
Normal	F	-	C2
Normal	F	92	C3
Normal	M	56	C4
Septum perforation	F	-	C5
Paediatric	F	7	C6
Paediatric	F	11	C7
Septum deviation	M	44	C8
Septum deviation	M	21	C9
Turbinate hypertrophy	F	23	C10
Turbinate hypertrophy	F	79	C11

**Table 2 pharmaceutics-15-02661-t002:** Factors used in the DoE of each anatomy.

Factors	Level (1)	Level (2)	Level (3)
A: Device	UDS	TriVair	Miat
B: Angle	Centre	Direct	-
C: Inspiratory flow (L/min)	0	15	30
D: Side	Left	Right	-

**Table 3 pharmaceutics-15-02661-t003:** Summary of the powder characteristics at the exit of each device.

Device	Dv50 (μm)	Span	Ejection Velocity (m/s)	Plume Angle (°)
UDS	25.21 ± 1.45	1.76 ± 0.03	49.20 ± 5.97	16.83 ± 2.42
TriVair	69.79 ± 12.20	1.89 ± 0.27	5.93 ± 0.14	3.93 ± 0.29
MIAT	38.24 ± 8.84	1.82 ± 0.37	7.49 ± 4.02	3.1 ± 0.5

**Table 4 pharmaceutics-15-02661-t004:** Summary of the anatomical measurements for each nasal cast.

Cast	Total Area of the Olfactory Region (mm^2^)	Total Area of the Nasal Valve (mm^2^)	Relative Difference between Left and Right Olfactory Areas (%)	Relative Difference between Left and Right Valve Areas (%)	Overall Volume of the Cavity (mm^3^)	Overall Area of the Cavity (mm^2^)	Width (mm)	Length (mm)	Height (mm)
C1	1334	84	-	-	17,226	10,220	15	125	48
C2	2200	213	7.76	33.96	56,637	26,331	49	108	69
C3	2274	123	3.88	**45.41**	57,949	27,395	50	104	67
C4	1082	226	9.44	4.31	84,385	31,109	55	114	40
C5	2200	213	7.76	33.96	56,637	26,331	49	108	69
C6	2158	98	43.63	18.73	19,063	16,854	32	94	36
C7	685	140	14.89	11.17	69,317	27,015	46	106	69
C8	1938	203	10.86	16.06	81,948	42,188	44	121	63
C9	3367	104	10.85	**42.29**	118,140	39,080	55	114	56
C10	1024	192	2.56	22.82	71,002	32,523	54	121	58
C11	723	149	8.62	21.56	38,852	27,378	55	107	44

**Table 5 pharmaceutics-15-02661-t005:** Maximal olfactory deposition and optimal parameters for each nasal cast. Angle are referred as “Direct aim” when the centre of the olfactory region and “Centre” when aiming at the centre of the internal nasal valve. Devices used are Aptar Unidose System (UDS), IPMed TriVair, and MIAT insufflator.

Cast	Mean Olfactory Deposition (%)	Maximal Olfactory Deposition (%)	Optimal Parameters to Reach the Olfactory Region				Maximal Deposition Predicted with Each Device (%)		
			Side	Device	Angle	Inspiratory Flow (L/min)	UDS	TriVair	MIAT
C1	19	35	-	UDS	Centre	0	33	13	11
C2	16	40	No influence	UDS	Direct aim	0	36	7	21
C3	12	43	Right	UDS	Centre	60	48	41	29
C4	7	29	Left	UDS	Direct aim	60	22	8	5
C5	17	48	Right	UDS	Direct aim	0	34	18	25
C6	29	59	No influence	UDS	no influence	No influence	47	24	20
C7	4	12	No influence	MIAT	Centre	0	9	6	11
C8	11	35	No influence	TriVair	Direct aim	60	25	31	16
C9	12	43	Left	UDS	Direct aim	60	42	18	21
C10	3	12	Right	UDS	no influence	60	17	15	12
C11	3	15	Left	TriVair	Direct aim	60	2	12	5

## Data Availability

The data presented in this study are available in the Appendix A. The scans are not publicly available due to privacy requirements.

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
