# Peer review of "What Are the Key Anatomical Features for the Success of Nose-to-Brain Delivery? A Study of Powder Deposition in 3D-Printed Nasal Casts"

_pharmaceutics, 2023, doi:10.3390/pharmaceutics15122661_

Round 1
Reviewer 1 Report
Comments and Suggestions for Authors
An interesting and promising work opening the way towards the utilization of the 3d printing technology and CT scanning in the field of nasal drug delivery to the brain. Such method will allow performing advance investigation of the mechanistic aspect of nasal direct nor to brain delivery.
However, the manuscript needs to be improved before being considered for publication.
Please see below my recommendations:
1. The title should be rewritten in a more specific manner focusing on the manuscript content. The current title gives impression that his is a review article about a wide topic. I suggest something like “Key anatomical features for the success of formulation deposition for nose-to- 2 brain delivery- a study on 3d Printer nasal casts”
The only studied parameter related to nose-to brain delivery is the formulation deposition. And this should be emphasized in the title, introduction, abstract and conclusion.
2. Abstract: the conclusion and future prospects are missing. These are very important parts to emphasize the importance of such work .
3. Introduction: a paragraph about the 3D printing technology related to the field of studying organ features. Are there any previous studies covering similar ideas?? Also please emphasize the promising prospects of such approach.
4. Materials and methods:
Section 2.1.2. the software used to translate the CT scan into chart for 3D printing is missing. The black resin should be specified, what is the material? Name, grade, manufacturer etc. Photos of the 3d printed nasal cavity parts should be given.
Section2.2.1. a table should be given to summarize the studied parameters
Line 155: As per my experience the maximum dose for nasal delivery is up to 100mg. Kindly check this detail. It is enough to write that the administrated amount is within the acceptable range for nasal administration.
5. Results: 3.3. line 265 what is this “Error! Reference source not found”
6. Discussion part is missing, maybe the authors want to address the results art as Results and Discussion.
7. Conclusions: too long. Again, future prospects are missing also what is the contribution of this work to the field???? Must be clearly stated.
Comments on the Quality of English Language
Minor editing of English language required
Reviewer 2 Report
Comments and Suggestions for Authors
The current manuscript is a quite interesting experimental study on the production of 3D models of the nasal cavity and assessment of powder deposition for nose-to-brain drug delivery purposes. Many relevant studies were performed, and both methodology and discussion seem sound. Hence, I only ask that the following changes are made before acceptance for publication:
- A figure regarding nose-to-brain drug delivery should be produced and added to the introduction section;
- Pictures of the assays should be added, especially of the produced 3D casts, since they will add a more visual component;
- It would be interesting to assess the deposition of liquid droplets, in addition to powders; can the authors comment on this, on if they intend to perform these assays in the future, and how it might differ from the results obtained in this study?;
- How these models might differ from the real case scenario, the biological human nasal cavity, should be commented;
- The novelty of the present work should be commented: has this work never been done before, by any other research group? I have found at least one review article on the subject (https://www.sciencedirect.com/science/article/pii/S0169409X21002167), and hence authors should compare their work to the one done by other authors before them, and discuss differences and similarities between methods and results;
- A section regarding statistical analysis should be added;
- Abbreviations should be explained in figure and table captions; an abbreviation list should also be added to the manuscript.
Round 2
Reviewer 1 Report
Comments and Suggestions for Authors
The authors made the the recomended imporvments
I suggest to accept this vervion of the manuscript
Comments on the Quality of English LanguageMinor editing of English language required